# One-shot preparation of topologically chimeric nanofibers via a gradient supramolecular copolymerization

Yuichi Kitamoto [1,5], Ziyan Pan[2,5], Deepak D. Prabhu[2], Atsushi Isobe[2], Tomonori Ohba [3], Nobutaka Shimizu [4], Hideaki Takagi[4], Rie Haruki[4], Shin-ichi Adachi [4] & Shiki Yagai [1,2]*

Supramolecular polymers have emerged in the last decade as highly accessible polymeric nanomaterials. An important step toward finely designed nanomaterials with versatile functions, such as those of natural proteins, is intricate topological control over their main chains. Herein, we report the facile one-shot preparation of supramolecular copolymers involving segregated secondary structures. By cooling non-polar solutions containing two monomers that individually afford helically folded and linearly extended secondary structures, we obtain unique nanofibers with coexisting distinct secondary structures. A spectroscopic analysis of the formation process of such topologically chimeric fibers reveals that the monomer composition varies gradually during the polymerization due to the formation of heteromeric hydrogen-bonded intermediates. We further demonstrate the folding of these chimeric fibers by light-induced deformation of the linearly extended segments.

[1] Institute for Global Prominent Research (IGPR), Chiba University, 1-33, Yayoi-cho, Inage-ku, Chiba 263-8522, Japan. [2] Division of Advanced Science and Engineering, Graduate School of Engineering, Chiba University, 1-33 Yayoi-cho, Inage-ku, Chiba 263-8522, Japan. [3] Department of Chemistry, Graduate School of Science, Chiba University, 1-33 Yayoi-cho, Inage-ku, Chiba 263-8522, Japan. [4] Photon Factory, Institute of Materials Structure Science, High Energy Accelerator Research Organization, Tsukuba 305-0801, Japan. [5] These authors contributed equally: Yuichi Kitamoto, Ziyan Pan. *email: yagai@faculty.chiba-u.jp

Programmed self-assembly of multicomponent molecular systems creates functional nanoarchitectures that comprises segregated supramolecular domains, and exhibit synergic combinations of different molecular properties[1–6]. Particularly in nanostructures with one-dimensional (1D) topologies, which can be prepared by the supramolecular polymerization of small molecules[7–9], such segregated supramolecular domains may exhibit intricate functions, such as domain-specific molecular recognition and conformational change. However, methods for the facile preparation of such programmed 1D topologies from a mixture of different molecules through spontaneous self-assembly remain challenging[10]. Although the segregated organization of molecular components into two-dimensional (2D) and three-dimensional (3D) nanostructures can be guided by differences in the molecular surface energy and interactions with substrates (Fig. 1a)[11–23], 1D nanostructures are generally prepared via self-assembly in solution, which allows individual molecular components to self-nucleate and elongate into self-sorted fibers (narcissistic self-sorting)[24–28]. Accordingly, in most cases, the synthesis of such multicomponent 1D nanostructures with segregated molecular domains, i.e., block supramolecular copolymers, relies on seeded-growth approaches, wherein kinetically inert seeds have to be prepared in advance from one monomer, whereas the other has to be co-assembled in a kinetically controlled manner in order to avoid self-nucleation[1–6,29–36]. Furthermore, as similar monomer structures are required to trick the feeded monomers into attaching to the termini of the seeds, structural complexity produced from the coexistence of distinct 1D topologies within single polymer chains, as seen in proteins, remains challenging.

Herein, we report the one-shot synthesis of topologically chimeric supramolecular copolymers that are comprised of distinct secondary-structure segments (Fig. 1b). This unique polymeric material is developed by the temperature-controlled supramolecular copolymerization of two different aromatic (naphthalene and anthracene) monomers, which individually afford supramolecular polymers with entirely different secondary structures, i.e., helically folded and linearly extended arrangements, despite their similar molecular structures. These monomers carry the same hydrogen-bonding heterocycle (barbiturate), which allows them to assemble into supermacrocycles (rosettes) that are capable of polymerizing primarily via π–π stacking interactions (Fig. 1c). Accordingly, these two monomers can potentially co-assemble into heteromeric rosettes. At the same time, the distinct aromatic surfaces of the two acene monomers enable self-sorting through π–π stacking interactions. We find that the interplay of these two supramolecular recognition events can be controlled kinetically, leading to a gradient supramolecular copolymerization, where the monomer composition varies gradually along the main chain. As such, we report the temperature-induced preparation of nanofiber topologies that comprise helically folded and linearly extended segments.

## Results

**Effect of the π-conjugated core on the secondary structure.** In a previous study, we showed that the barbiturate-derived molecule Nap, which contains a diphenylnaphthalene core, affords helically folded supramolecular polymers upon slowly cooling (1.0 K min⁻¹) a hot methylcyclohexane (MCH) solution (Fig. 2a, e)[37]. The helical folding is due to the specific π–π stacking arrangement of the six-membered hydrogen-bonded rosettes[38–40] with rotational and translational offsets, which generates an intrinsic curvature in the main chain with a curvature radius of ca. 13 nm (Fig. 2c)[41–43]. To our surprise, the modification of the π-conjugated core from naphthalene to anthracene (Nap → Ant; Fig. 2b) resulted in an entirely different morphology, i.e., linearly extended fibers lacking intrinsic curvature, as revealed by atomic force microscopy (AFM) and transmission electron microscopy (TEM) (Fig. 2f and Supplementary Fig. 1). The small-angle X-ray scattering (SAXS) profile of the Ant solution showed only fractal-like scattering, which is very different from that of Nap (Supplementary Fig. 2)[37]. The latter displays a non-periodic oscillatory feature in the range $Q = 0.3$–$0.9$ nm⁻¹, which we attributed to its intrinsic curvature[40]. The width of the Ant fiber (ca. 8.6 nm; Fig. 2f) is by 0.8 nm larger than that of Nap (ca. 7.8 nm; Fig. 2e), which is consistent with the difference of the diameters of their rosettes (Fig. 2a, b). Neither very slow cooling (0.1 K min⁻¹) nor prolonged aging of the solution (several weeks) induced any curvature on the Ant fibers. Force-field molecular mechanics calculations of the Ant and Nap rosettes revealed almost the same geometrical features in terms of twisting of the acene moieties, while upon stacking the rosettes, a larger overlap of the acene moieties was observed for Ant (Supplementary Figs. 3 and 4). Indeed, upon cooling, the Ant solution showed merely a simple hypsochromic shift of its broad absorption band (Fig. 2h), which stands in contrast to the bathochromic shift

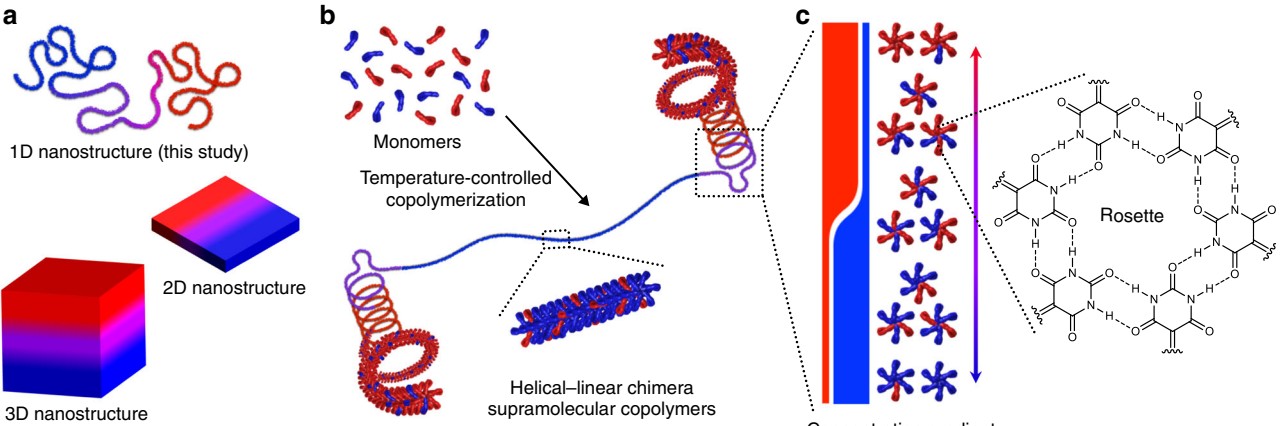

**Fig. 1** Schematic illustration of supramolecular nanofibers that are comprised of nanosegregated secondary-structure motifs. **a** Diagrams of 1D, 2D, and 3D nanomaterials with a concentration gradient of their constituent molecules. **b** Schematic representation of the spontaneous formation of supramolecular copolymers with helical–linear chimeric topology from a mixture of two essentially self-sorting monomers (colored in red and blue) via the formation of hydrogen-bonded supramolecular intermediates (rosettes). **c** Diagram of the internal concentration gradient of red and blue monomers, which enables alternation of the major components responsible for the distinct structural motifs during continuous growth of the supramolecular backbone

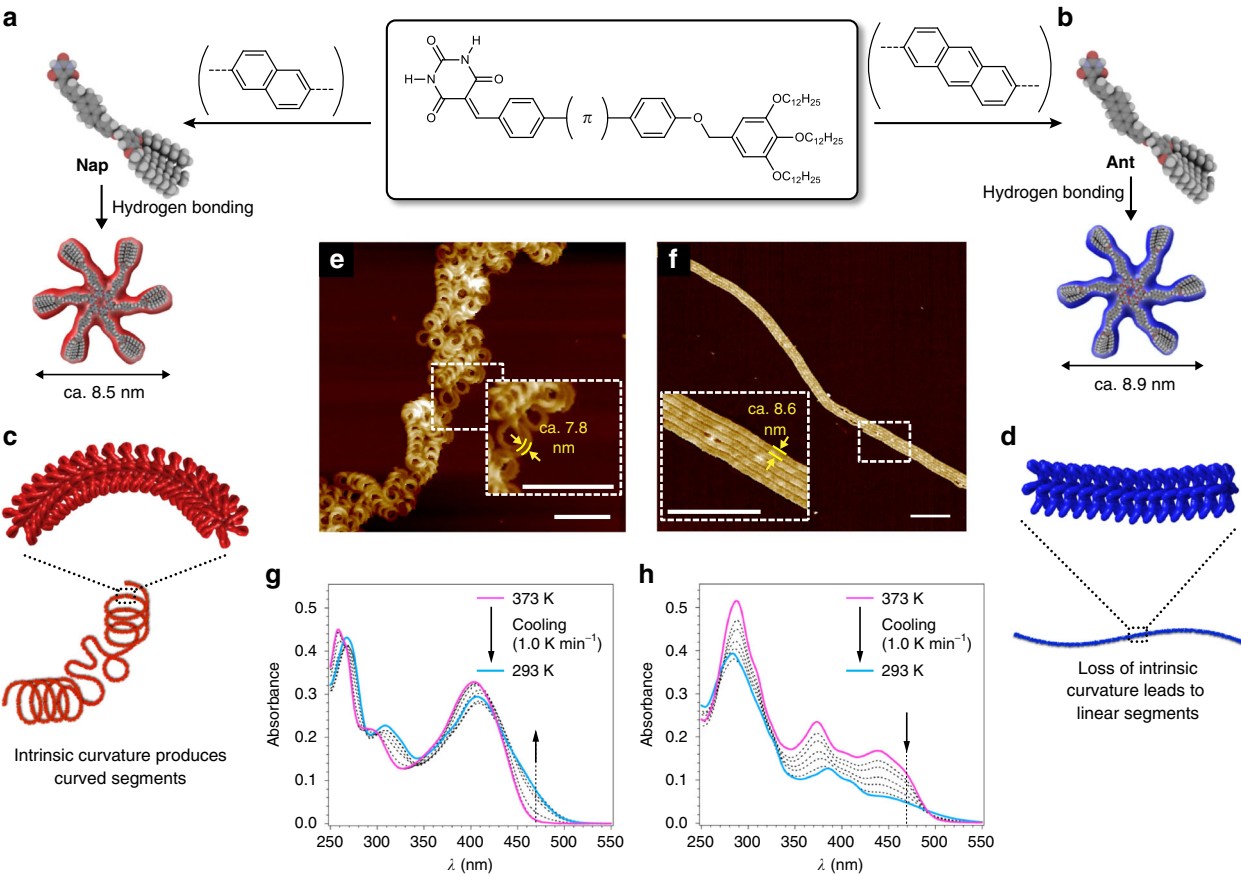

**Fig. 2** Impact of the π-conjugated core on the secondary structures of the supramolecular polymers. **a, b** General molecular structure and hydrogen-bonded hexamers (rosettes) of Nap (**a**) and Ant (**b**) with their CPK molecular models. **c, d** Schematic representation of the supramolecuar polymerization of Nap (**c**) and Ant (**d**) into helically coiled and linearly extended secondary structures, respectively. **e, f**, AFM images of supramolecular polymers of Nap (**e**) and Ant (**f**) prepared by cooling a hot MCH solution of each molecule ($c = 1.0 \times 10^{-5}$ M) from 373 to 293 K using a cooling rate of 1.0 K min$^{-1}$. The inset images show a magnified image of the area enclosed by the dashed rectangles. Scale bars, 100 nm. **g, h** Temperature-dependent ultraviolet–visible (UV–Vis) absorption spectra of Nap (**g**) and Ant (**h**) in MCH ($c = 1.0 \times 10^{-5}$ M) upon cooling from 373 to 293 K using a cooling rate of 1.0 K min$^{-1}$

observed for that of Nap (Fig. 2g). Accordingly, the extension of the acene moiety from naphthalene to anthracene results in a smaller degree of rotational/translational displacement of the stacked rosettes due to stronger π–π stacking interactions, which could be a major cause for the loss of intrinsic curvature (Fig. 2d). These dramatically different supramolecular homopolymer topologies for Nap and Ant, despite their similar molecular structures, motivated us to explore one-shot preparations[44–49] of supramolecular copolymers, where the two components are inhomogeneously mixed to form segments with distinct secondary structures.

**Kinetic preparation of chimeric supramolecular fibers**. We studied the temperature-induced supramolecular copolymerization of Ant and Nap by cooling a monomeric mixture in MCH from 373 to 293 K at a constant cooling rate. As discussed later, we chose 1:1.5 as the optimum mixing ratio for Ant ($c = 1.0 \times 10^{-5}$ M) and Nap ($c = 1.5 \times 10^{-5}$ M). We discovered that the cooling rate, which greatly affects degree of kinetic contribution in surpamolecular polymerization[43,50–54], is also a key parameter for such one-shot gradient supramolecular copolymerizations. For example, a very slow cooling rate (0.1 K min$^{-1}$) resulted in a simple mixture of linearly extended fibers and coiled fibers, as shown by AFM (Fig. 3a and Supplementary Fig. 5a, b). This result indicates that under thermodynamic conditions, the two monomers are allowed to self-sort. On the other hand, nanofibers comprising helically coiled and linearly extended segments were obtained upon applying

higher cooling rates (1.0–5.0 K min$^{-1}$; Fig. 3b and Supplementary Fig. 5c, d). Most fibers present an ABA-(helical–linear–helical)-type structure, which suggests that Ant first forms a linear segment, before Nap forms helical segments (vide infra). Hereafter, we refer to these unique fibers, which feature distinct secondary-structure segments, as chimeric fibers. It should be noted that these chimeric fibers are kinetically very stable, as their population in solution remains unchanged, even after 1 week.

While the fusion of two well-defined secondary-structure segments was confirmed unambiguously by AFM measurements, the interplay of the supramolecular polymerization processes of the two monomers was examined by variable-temperature absorption measurements (Fig. 3c, d). To analyze these spectra, we also evaluated the individual supramolecular polymerization processes in MCH. As already shown in Fig. 2g, h, upon cooling from 373 (fully monomeric state) to 293 K (fully aggregated state), both Nap and Ant display absorption changes that should be attributed to electronic interactions between the aromatic cores. By plotting the absorbance change ($\Delta A = A_T - A_{373}$) at 470 nm as a function of the temperature, we obtained contrasting curves, i.e., $\Delta A$ increases negatively for Ant and positively for Nap, which results from their different stacking modes (blue and red curves in Fig. 3e, f). Both curves are non-sigmoidal, suggesting a cooperative supramolecular polymerization that is characterized by the elongation temperature ($T_e$) that separates the nucleation and elongation processes (Supplementary Fig. 6a, b). The $T_e$ was found to increase with increasing initial concentration of the monomers

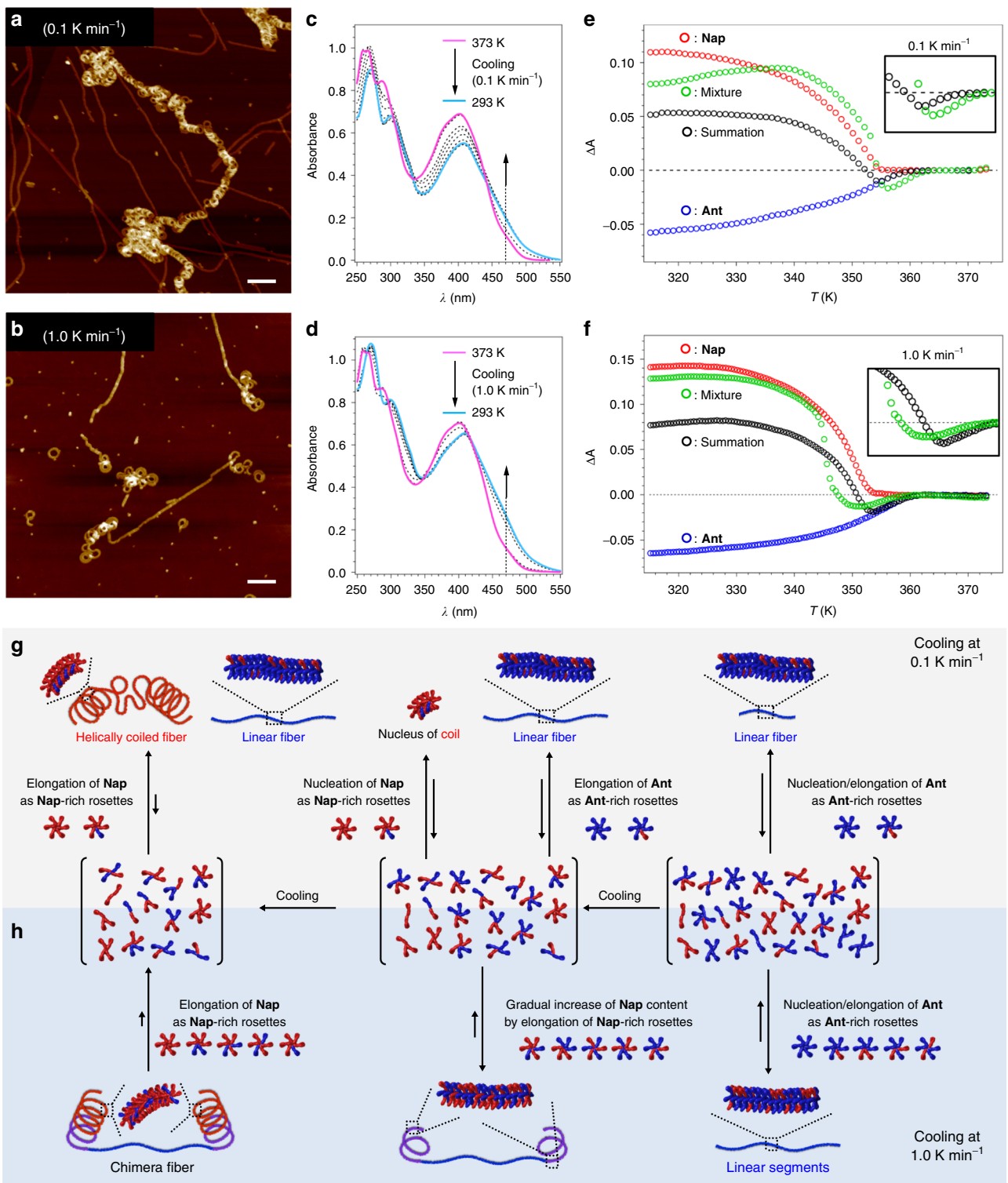

**Fig. 3** Cooling-rate-dependence of the gradient supramolecular copolymerization. **a**, **b** AFM images of supramolecular (co)polymers prepared by cooling a hot MCH solution of a 1:1.5 mixture of Ant ($c = 1.0 \times 10^{-5}$ M) and Nap ($c = 1.5 \times 10^{-5}$ M) from 373 to 293 K using cooling rates of 0.1 K min$^{-1}$ (**a**) and 1.0 K min$^{-1}$ (**b**). Scale bars, 100 nm. **c**, **d** Temperature-dependent UV–Vis absorption spectra of a 1:1.5 mixture of Ant ($c = 1.0 \times 10^{-5}$ M) and Nap ($c = 1.5 \times 10^{-5}$ M) upon cooling from 373 to 293 K using cooling rates of 0.1 K min$^{-1}$ (**c**) and 1.0 K min$^{-1}$ (**d**). **e**, **f** Plot of the absorption changes ($\Delta A = \Delta A_T - \Delta A_{373}$) at $\lambda = 470$ nm as a function of temperature for Ant (blue circles, $c = 1.0 \times 10^{-5}$ M), Nap (red circles, $c = 1.5 \times 10^{-5}$ M), and a 1:1.5 mixture of Nap and Ant (green circles, [Ant] $= 1.0 \times 10^{-5}$ M; [Nap] $= 1.5 \times 10^{-5}$ M) in MCH upon cooling from 373 to 293 K using cooling rates of 0.1 K min$^{-1}$ (**e**) and 1.0 K min$^{-1}$ (**f**). The black circles correspond to a reference plot obtained by a simple summation of the $\Delta A$ values of the two homoassemblies. **g**, **h** Schematic representation of the plausible process underlying the gradient supramolecular copolymerization that affords a simple mixture of linearly extended fibers and coiled fibers by 0.1 K min$^{-1}$ cooling (**g**) and of chimeric fibers by 1.0 K min$^{-1}$ cooling (**h**). The blue and red monomers represent Ant and Nap, respectively

(Supplementary Fig. 6c). The $T_e$ of Ant at $c = 1.0 \times 10^{-5}$ M (361 K) is higher than that of Nap at $c = 1.5 \times 10^{-5}$ M (353 K). Although the two monomers in a mixture at this ratio may form an equilibrium mixture of diverse hydrogen-bonded species before nucleation, Ant rosettes can nucleate first owing to it's highest aggregation capability, and Ant is further consumed from the mixture through self-recognition process driven by π−π stacking interactions. Accordingly, the experimentally observed helical–linear–helical structure reflects well the higher $T_e$ of Ant than that of Nap. The sum of the above homopolymerization curves provides a characteristic down–up trajectory with decreasing temperature (black curve in Fig. 3e, f), pointing to a higher $T_e$ for Ant than for Nap in a 1:1.5 mixture.

These simulated curves emulate a situation wherein the individual polymerization pathways of the two monomers in the 1:1.5 mixture do not interplay but lead to narcissistic self-sorting. The most important feature of these calculated curves is the sharp negative-to-positive transition of $\Delta A$ at 354 K, which indicates the nucleation of Nap during its cooperative supramolecular polymerization. Indeed, the experimental curve recorded at a cooling rate of 0.1 K min$^{-1}$ (green curve, Fig. 3e), for which self-sorted fibers were observed by AFM (Fig. 3a), exhibits a similarly sharp $\Delta A$ transition (Fig. 3e, inset). The transition temperature of the experimental curve is higher by 2.5 K than that of the simulated curve. This is most likely because a minor amount of Nap is incorporated as Ant-rich rosettes (e.g., rosettes composed of five Ant and one Nap molecules). If such Ant-rich rosettes are allowed to nucleate with the homomeric rosette of Ant, the nucleation temperature should be higher than that of pure Ant because of an apparent increase of the concentration of Ant, which is in line with the experimental data. In the elongation regime, on the other hand, the $\Delta A$ values of the experimental curve are always larger than those of the simulated curve (Fig. 3a). This can be explained in line with the incorporation of a minor amount of Ant in Nap-rich rosettes. The incorporated Ant molecules should also form slipped stacking to show bathochromic shift, contributing on the increase of $\Delta A$ at 470 nm. Although the two monomers cannot self-recognize through hydrogen-bonding and form an equilibrium mixture of diverse hydrogen-bonded species before nucleation, they can quasi-narcissistically self-sort into Ant-rich and Nap-rich fibers under thermodynamic conditions (Fig. 3g). The experimental and simulated absorption spectra at a temperature slightly lower than the nucleation temperature of Ant are exactly the same (Supplementary Fig. 7), which further corroborates that Ant first nucleate from the equilibrium mixture.

In contrast, the experimental curve recorded at a cooling rate of 1.0 K min$^{-1}$ showed a moderate negative-to-positive transition of $\Delta A$ in the range of 352–348 K (green curve, Fig. 3f). The moderate increase in the number of aggregated Nap molecules is indicative of nucleation suppression, which strongly suggests that Nap can be gradually incorporated into elongating Ant-rich fibers[55–59] under kinetic conditions that do not allow their self-recognition through π−π stacking interactions (Fig. 3h). In this sense, we have realized a gradient supramolecular copolymerization.

The chimeric structural features of the supramolecular copolymers were further evaluated by fluorescence studies. Owing to the distinct stacking mode of the rosettes, Ant and Nap display distinct emission properties in the aggregated state (Supplementary Fig. 8). At 373 K, both monomers exhibit vibronic emission bands that should be attributed to the emission from a locally excited (LE) state according to density functional theory (DFT) calculations (Supplementary Fig. 9). Upon cooling, only Nap shows a new broad emission band that is shifted toward the visible region, whereas a simple increase of the LE emission is

observed for Ant. The broad emission band of Nap can be interpreted in terms of excimer emission derived from slipped stacking of the naphthalene moieties (Supplementary Fig. 4). The absence of a similar emission band for Ant, on the other hand, can be rationalized by a larger overlap of the anthracene moieties, which does not allow radiative excited state[60]. Instead, the LE emission is enhanced as a result of suppression of non-radiative decay such as bond twist by aggregation. While the excimer emission of Nap and the LE emission of Ant are sufficiently separated, the absorption band of Nap overlaps considerably with the LE emission of Ant (Fig. 4a)—a situation where energy transfer from Ant to Nap may occur if they are co-assembled in supramolecular polymers. At 293 K, when the solution of the 1:1.5 mixture prepared by cooling at 1.0 K min$^{-1}$ was excited at 283 nm, i.e., where Ant is preferentially excited, the LE emission intensity of Ant ($\lambda_{em} = 426$ nm) was attenuated by a factor of 0.62, while the excimer emission of Nap ($\lambda_{em} = 557$ nm) was enhanced by a factor of 1.5 in comparison with those of a mixture of preformed homopolymers (Supplementary Fig. 10). The same difference could be confirmed when compared to the cooling at 0.1 K min$^{-1}$ with a factor of 0.68 for the attenuation of the LE emission from Ant and a factor of 1.34 for the enhancement of the excimer emission from Nap (Fig. 4b, c). This difference subsequently manifests in different emission colors (Fig. 4d).

We further monitored the emission intensity of Ant upon cooling a 1:1.5 mixture at 0.1 or 1.0 K min$^{-1}$. In case of 0.1 K min$^{-1}$, the Ant emission in the mixture increases monotonously, similarly to the case of Ant homopolymerization (blue curves in Fig. 4f). This clearly reflects the orthogonal self-assembly processes of Ant and Nap. On the other hand, in the case of 1.0 K min$^{-1}$, the Ant emission in the mixture temporarily decreases from 350 to 340 K (blue curves in Fig. 4e), which was then compensated by a temporarily sharp increase of the Nap emission (red curves in Fig. 4e). Similar results were obtained at a cooling rate of 5.0 K min$^{-1}$ (Supplementary Fig. 11). These results demonstrate the interplay of the supramolecular polymerization processes of the two monomers under kinetic conditions.

**Modulation of the chimeric structure.** Another important factor during the gradient supramolecular copolymerization is the concentration of the respective monomers, which is reflected in the fact that unbalanced mixing failed to yield gradient supramolecular copolymers. For example, 1:0.5 mixing of Ant ($c = 1.0 \times 10^{-5}$ M) and Nap ($c = 0.5 \times 10^{-5}$ M) resulted in a mixture of extended linear fibers characteristic of Ant and a minor amount of short curved fibers (Fig. 5a and Supplementary Fig. 12), even though Nap itself can form helically elongated fibers at this low concentration[37]. Conversely, when Nap was added in excess to Ant (e.g., 1:2, 1:2.5, and 1:3 mixtures), only the helically or randomly coiled fibers characteristic of Nap were observed (Fig. 5b and Supplementary Figs. 13–15). These results suggest that, if one monomer is present in large excess[61], self-nucleation of the minor monomer does not occur, and the heteromeric rosettes that are primarily composed of the major monomer organize into secondary structures characteristic of the major monomer component without forming intermediate topologies featuring larger curvature radii.

As such, one-shot gradient supramolecular copolymerization occurs only when the two monomers are mixed in a balanced fashion so that they can essentially self-sort (e.g., Ant:Nap = 1:1–1:1.5). To quantitatively analyze the yield of the copolymers, we randomly captured more than ten AFM images of a sample solution, and analyzed all existing fibers in the acquired images using a graphic software program (Supplementary Method). By

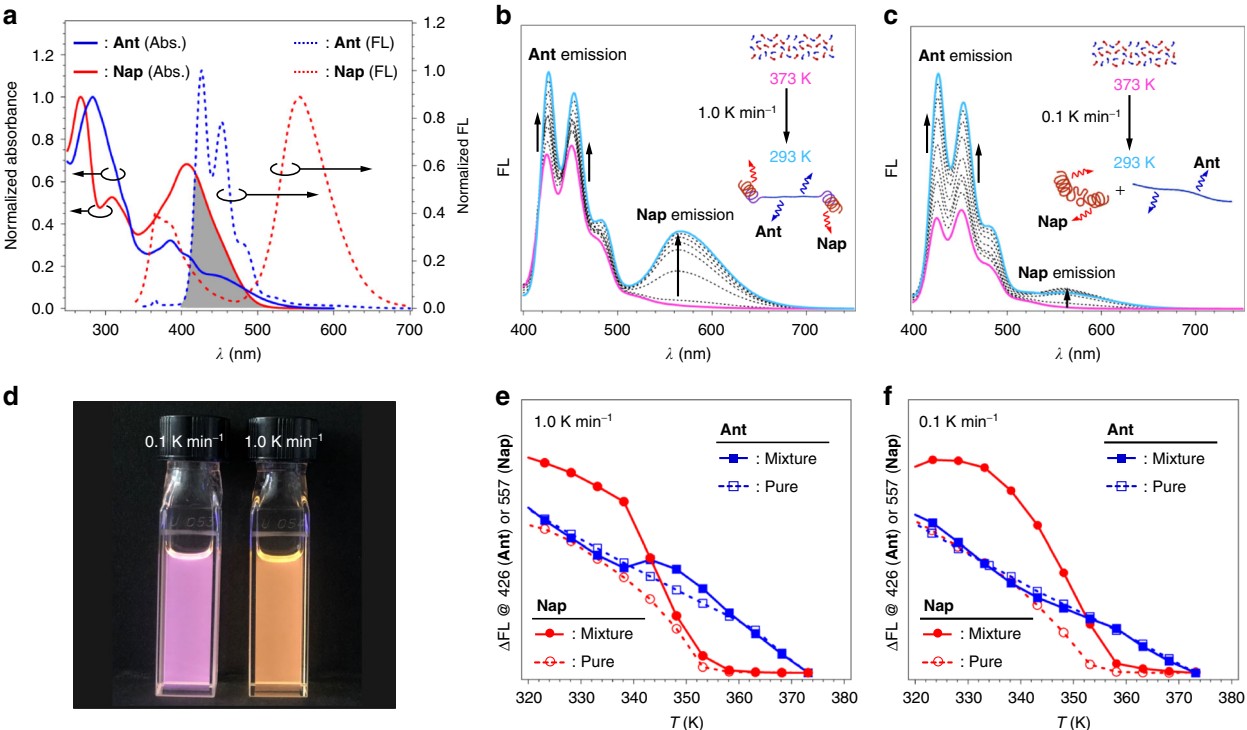

**Fig. 4** Mechanistic studies on the gradient supramolecular copolymerization by fluorescence spectroscopy. **a** Overlay of fluorescence spectra and absorption spectra of Ant and Nap ($c = 1.0 \times 10^{-5}$ M). Dotted curves: fluorescence spectra of Ant (blue) and Nap (red) in MCH ($\lambda_{ex} = 283$ nm) measured after cooling from 373 to 293 K using a cooling rate of 1.0 K min$^{-1}$. Solid curves: absorption spectra of Ant (blue) and Nap (red) in MCH measured after cooling from 373 to 293 K using a cooling rate of 1.0 K min$^{-1}$. The gray area shows the overlap of the fluorescence spectrum of Ant and absorption spectrum of Nap. **b**, **c** Temperature-dependent fluorescence spectra of a 1:1.5 mixture of Ant ($c = 1.0 \times 10^{-5}$ M) and Nap ($c = 1.5 \times 10^{-5}$ M) upon cooling from 373 to 293 K using cooling rates of 1.0 K min$^{-1}$ (**b**) and 0.1 K min$^{-1}$ (**c**) ($\lambda_{ex} = 330$ nm). **d** Photograph of the emission of 1:1.5 mixtures prepared by cooling at 0.1 K min$^{-1}$ (left) and 1.0 K min$^{-1}$ (right). **e**, **f** Plot of the fluorescence intensity change ($\Delta$FL) for Ant at 426 nm (blue) and Nap at 557 nm (red) as a function of the temperature upon cooling pure MCH solutions (open circles and squares with dashed lines) and the 1:1.5 mixure (closed circles and squares with solid lines) using a cooling rate of 1.0 K min$^{-1}$ (**e**) and 0.1 K min$^{-1}$ (**f**)

virtue of the distinct secondary structures of Ant and Nap, we could clearly discriminate copolymers from homopolymers. From the 1:1 mixture ([Ant] = [Nap] = $1.0 \times 10^{-5}$ M), only a small amount of copolymers was observed. Among the 173 fibers found in the 14 AFM images recorded, 35 (20%) and 126 (73%) fibers were composed solely of linearly extended and helically coiled secondary structures, respectively, suggesting that they were homopolymers with minor inclusion of the other monomer (Fig. 5k). Only the remaining 12 (7%) fibers presented chimeric structures (Fig. 5k), but these were predominantly composed of a linear segment with spirally curled ends (Fig. 5c and Supplementary Fig. 16). The average length ratio of the linear and coiled segments of these chimeric fibers ($L_{linear}:L_{coil}$) obtained from the 1:1 mixture was estimated as 60:40. The low yield of the chimeric fibers and their predominant linear segments that are primarily composed of pure Ant and Ant-rich rosettes can be correlated with the smaller overlap of the temperature regime for polymerizing pure Ant and Nap at $1.0 \times 10^{-5}$ M: the molar fraction of aggregated Ant molecules already reaches 0.65 at the $T_e$ (345 K) of Nap (Supplementary Fig. 6d).

A larger overlap of the temperature regime of polymerization was achieved for the concentration of Ant at $1.0 \times 10^{-5}$ M by increasing the concentration of Nap to e.g. $1.3 \times 10^{-5}$ M (1:1.3 mixture) and $1.5 \times 10^{-5}$ M (1:1.5 mixture), where the molar fraction of aggregated Ant molecules at the $T_e$ (350 and 353 K) of Nap was reduced to 0.5 and 0.4, respectively (Supplementary Fig. 6d). Under these conditions, we obtained ca. 30% and 45% of fibers as chimeric structures featuring helically coiled ends from the 1:1.3 and 1:1.5 mixtures, respectively (Fig. 5d–k and

Supplementary Figs. 17–19). These chimeric fibers showed an elongation of the coiled segments with average $L_{linear}:L_{coil} = 54:46$ and 37:63 for the 1:1.3 and 1:1.5 mixtures, respectively. From the $L$ values, degree of polymerization (DP) for each segment was calculated (Supplementary Table 1), giving $DP_{linear}:DP_{coil} = 1:0.9$ and 1:1.8 for 1:1.3 and 1:1.5 mixtures, respectively. This result shows the proportion of $DP_{coil}$ in the chimeric fibers increase by increasing the concentration of Nap. Accordingly, the percentage of chimeric fibers, whose coiled segments account for 50% of the total fiber length, increased from 22% in the 1:1 mixture to 46% in the 1:1.3 mixture, and to 79% in the 1:1.5 mixture (Fig. 5l–n). It should be noted that the respective segments are not purely composed of one monomer, but contain a minor amount of the co-monomer, as already demonstrated by the experiments for the largely biased mixtures.

**Photoinduced folding.** To shed light onto the quasi-block nature of our gradient supramolecular copolymers, we attempted segment-selective topological alterations by photochemical reactions. In the present system, Ant was expected to undergo intermolecular photodimerization[62,63] since its anthracene cores overlap significantly upon the supramolecular polymerization (Supplementary Fig. 4b, d). We thus initially examined the possibility of photoinduced topological changes in the supramolecular homopolymer of Ant. Upon irradiation of the homopolymer solution with ultraviolet (UV) light ($\lambda = 365$ nm; 17 W LED lamp) for 90 min, attenuation of the absorption and emission bands was observed, which is characteristic for the

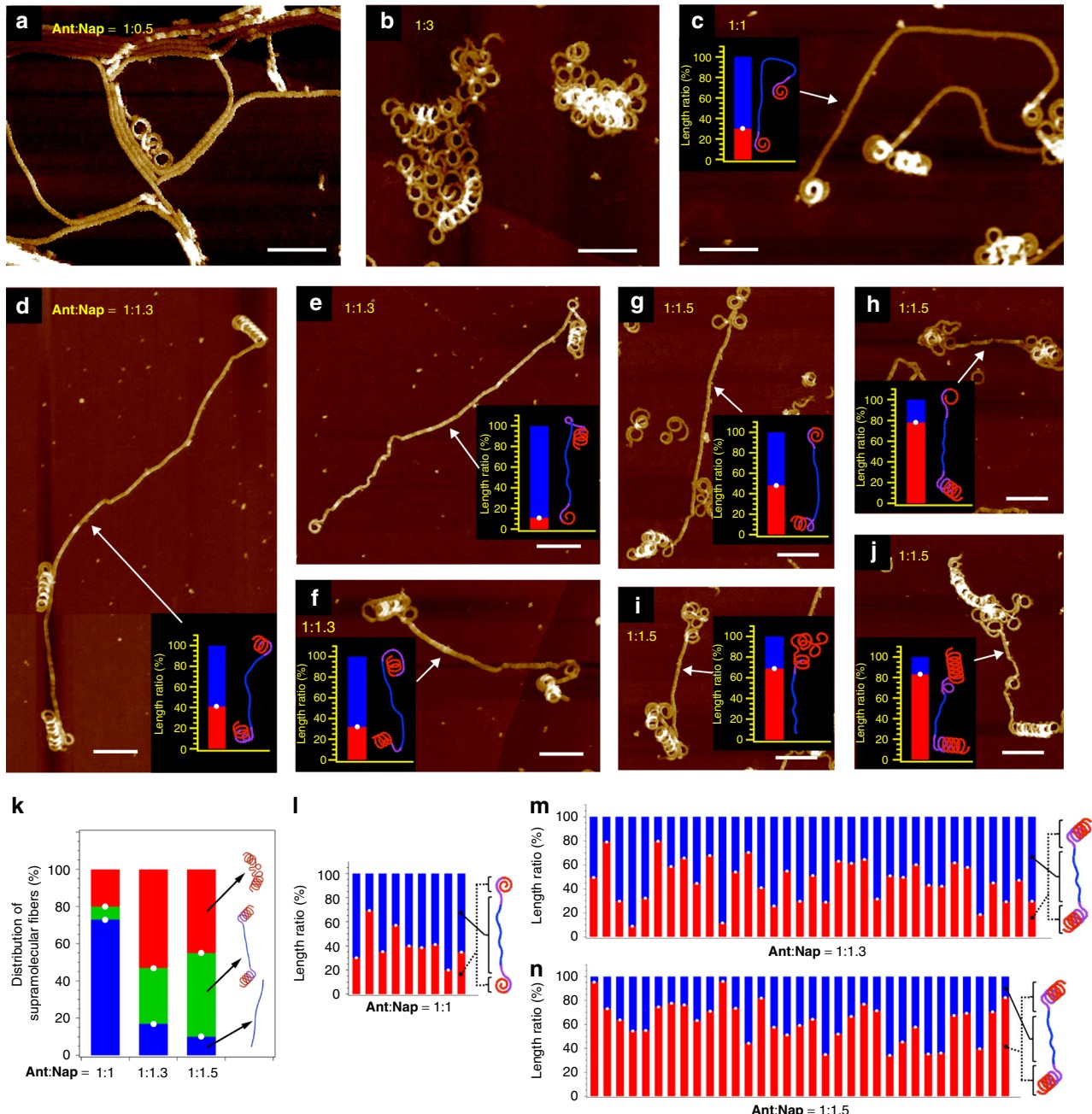

**Fig. 5** Structural analysis of the supramolecular copolymers. **a–j** AFM images of supramolecular copolymers prepared by cooling hot MCH solutions of 1:0.5 (**a**), 1:3 (**b**), 1:1 (**c**), 1:1.3 (**d–f**), and 1:1.5 (**g–j**) mixtures of Ant and Nap from 373 to 293 K using a cooling rate of 1.0 K min⁻¹. The concentration of Ant in all the mixtures remained constant ($1.0 \times 10^{-5}$ M), whereas that of Nap varied according to the mixing ratio (scale bars, 100 nm) Inset bar graphs show the length ratio of the linear (blue) and coiled segments (red). **k** Distribution of supramolecular fibers with helical (red), chimera (green) and linear (blue) topologies for the 1:1, 1:1.3, and 1:1.5 mixtures of Ant and Nap from the analysis of 14, 22, and 12 AFM images, respectively. The total number of fibers was 173, 158, and 125 for the 1:1, 1:1.3, and 1:1.5 mixtures, respectively. **l–n** Length ratio of the linear and curved segments in each chimera fiber for the 1:1 (**l**), 1:1.3 (**m**), and 1:1.5 (**n**) mixtures of Ant and Nap

photoreaction of anthracene chromophores (Supplementary Fig. 20a, b). Based on nuclear magnetic resonance (NMR) and UV–visible (Vis) analyses in CDCl₃ (Supplementary Fig. 20c–e), a photoconversion yield of 16% in MCH was established. Despite the low photoconversion yield, AFM measurements revealed inhomogeneously curved UV-irradiated Ant fibers (Fig. 6a and Supplementary Fig. 21). The inhomogeneity of the curvature was corroborated by a marginal change in the SAXS profile (Supplementary Fig. 22). The UV-induced curvature can thus be attributed to an attenuation of the stiffness of the supramolecular

polymer chains due to defects randomly generated upon photo-dimerization (Fig. 6b). In sharp contrast, the helical fibers of Nap were virtually insensitive to the same UV-light treatment, and significant changes were not observed for the absorption or emission spectra (Supplementary Fig. 23). These results are not surprising considering the intrinsically low photoreactivity of naphthalene chromophores and their smaller overlap upon supramolecular polymerization.

UV irradiation of a chimeric fiber solution (Ant:Nap = 1:1.5) for 90 min resulted in a comparable emission decrease for Ant,

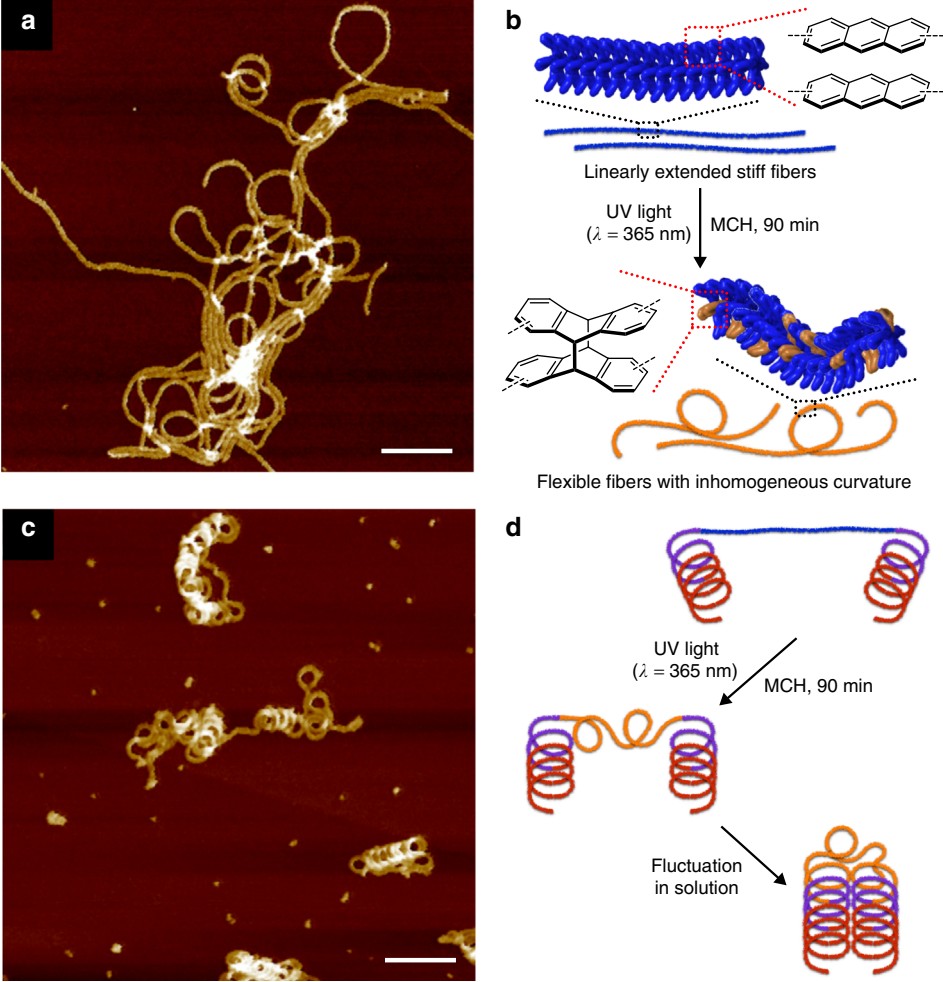

**Fig. 6** UV-induced morphology changes in Ant fibers and chimeric fibers. **a** AFM image of Ant fibers after 90 min of irradiation with UV light ($\lambda = 365$ nm) in MCH at 293 K. The original linear fibers were prepared by cooling a hot MCH solution of Ant ($c = 1.0 \times 10^{-5}$ M) from 373 to 293 K using a cooling rate of 1.0 K min$^{-1}$ (scale bar, 100 nm). **b** Schematic representation of the topology changes upon UV irradiation of Ant fibers. **c** AFM image of chimeric fibers after irradiation with UV light ($\lambda = 365$ nm) in MCH at 293 K for 90 min. The original chimeric fibers were prepared by cooling a hot MCH solution of a 1:1.5 mixture of Ant ($c = 1.0 \times 10^{-5}$ M) and Nap ($c = 1.5 \times 10^{-5}$ M) from 373 to 293 K using a cooling rate of 1.0 K min$^{-1}$ (scale bar, 100 nm). **d** Schematic representation of the topology changes upon UV irradiation of chimeric fibers

again suggesting photodimerization (Supplementary Fig. 24a). In contrast to the homopolymer case, the emission of Nap was also reduced after UV irradiation (Supplementary Fig. 24b). As the SAXS profile of the solution remained unchanged after UV irradiation (Supplementary Fig. 24c), the reduction of the Nap emission can be most likely ascribed to the decrease in the concentration of Ant due to the photodimerization. The AFM images of the UV-irradiated mixture show compactly folded fibers by the loss of Ant-rich linear segments, while the Nap-rich helical segments remained intact (Fig. 6c and Supplementary Fig. 25). This dramatic topology changes can be rationally interpreted by the loss of stiffness of the Ant-rich linear segments after the photoreaction, allowing intrachain cohesion of Nap-rich helical segments[37] through dynamic conformational changes in solution (Fig. 6d).

## Discussion

We have demonstrated a supramolecular polymerization concept that provides access to unique nanostructures. The non-covalent polymerization of two monomers that inherently exhibit distinct nanofiber topologies afforded supramolecular nanofibers with segregated helical–linear secondary-structure segments. Remarkably,

this procedure does not require the prior preparation of any segments, which is often the case with supramolecular block copolymers[1–6,29–36]. Instead, a simple mixing of the two monomers under a temperature-gradient regime results in their spontaneous segregation in a single fiber owing to the different binding strengths of the respective monomers. This situation should, in principle, afford self-sorted supramolecular homopolymers. However, the hierarchical nature of our supramolecular polymerization approach, i.e., the indirect polymerization of monomers via the formation of hydrogen-bonded supermacrocyclic intermediates, and the very high kinetic stability of the main chains thus formed, leads to a gradual variation of the monomer composition along the main chain. This is a remarkable merit of self-assembled small molecular materials with well-programmed hierarchical nature, and can potentially be applied to other materials, such as liquid crystals and gels. In addition to this methodology, the level of topological control on the present supramolecular copolymers represents another important step toward the realization of complex functional properties, similar to those exhibited by proteins, using supramolecular polymer materials. Toward this goal, we are currently exploring the supramolecular copolymerization of further functional building blocks that are active in metal-coordination and/or chemical reactions.

## Methods

**Materials**. Compound Ant was synthesized according to the section of synthesis in Supplementary materials. Nap was synthesized according to our previous report[37]. All starting materials and reagents were purchased from commercial suppliers and used without further purification. Air sensitive reactions were conducted under nitrogen atmosphere using dry solvents. For spectroscopic measurements, spectroscopic grade solvents were employed. $^1H$ nuclear magnetic resonance (NMR; Supplementary Fig. 26a), $^{13}C$ NMR (Supplementary Fig. 26b), and electrospray ionization high-resolution mass (ESI-HRMS) spectra details of Ant are included in Supplementary materials.

**Atomic force microscopy**. AFM imaging was performed under ambient conditions using Multimode 8 Nanoscope V (Bruker Instrument) in Peak Force Tapping (ScanAsyst) mode. Silicon cantilevers (SCANASYST-AIR) with a spring constant of 0.4 N/m and frequency of 70 kHz (nominal value, Bruker, Japan) were used. The sample was prepared by spin-coating (3000 rpm, for 1 min) the MCH solution of supramolecular polymers (10 µl) onto freshly cleaved highly-oriented pyrolytic graphite (HOPG). Images were processed using NanoScope Analysis 1.40 (Bruker).

**Analysis of nanofiber lengths captured by AFM**. The lengths of supramolecular nanofibers were measured by using ImageJ software program (version 1.52a, National Institutes of Health, United States)[64] as follows. We initially processed the AFM images of nanofibers captured by AFM measurements by using ImageJ, and then set the range to fix the scale, e.g., 500 nm, 1 µm. The line of fibers can be traced manually to derive the length of the fiber based on the scale.

## Data availability

All data needed to evaluate the conclusions in the paper are present in the paper and its Supplementary Information files. Additional data related to this paper may be requested from the authors.

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

## Acknowledgements

This work was supported by KAKENHI grant no. 26102010 and a Grant-in-Aid for Scientific Research on Innovative Areas π-Figuration (grant no. 26102001) from the Japanese Ministry of Education, Culture, Sports, Science, and Technology (MEXT). This work was performed with the approval of the Photon Factory Program Advisory Committee (proposal no. 2016G550). S.Y. acknowledges financial support from Sekisui Integrated Research. We also thank Prof. Masami Kamigaito (Nagoya University) for fruitful discussion.

## Author contributions

S.Y., Y.K. and Z.P. designed the project. Z.P. performed the experimental work, except for the sysnthesis of Ant and Nap. D.D.P. synthesized Ant and Nap, while N.S., H.T., R.H. and S.-i.A. collected the SAXS data. T.O. performed the TEM experiments. S.Y. and Y.K. wrote the manuscript, while S.Y. and Y.K. prepared the figures. All authors including A.I. have contributed by commenting on the manuscript. The overall project was directed by S.Y.

## Competing interests

The authors declare no competing interests.
