## [Transparent Peer Review File · Nature Communications]

Reviewers' comments:

Reviewer #1 (Remarks to the Author):

In this manuscript, the authors report formation of topologically unique 'chimeric' nanofiber. Such a complex structure has not been achieved before, and this beautiful demonstration will surely lead to design of nanostructures such as those of proteins as the authors aim at. The work deserves publication in Nature Communications. I hope that the authors address the following comments before publication.

I agree that the hierarchical nature of supramolecular polymerization via the formation of hydrogen-bonded intermediate is crucial. I think the proposed mechanism is qualitatively correct, but as the authors admit by describing that "It should be noted that the respective segments are not purely composed of one monomer, but contain a minor amount of the co-monomer, as already demonstrated by the experiments for the largely biased mixtures (page 12)", the system is complex. The authors sought to quantitatively discuss the mechanism, but it has made discussion confusing (or misleading). I hope that the authors can estimate statistic proportions of each rosette (13 patterns) in the mixture before the nucleation. In the mixture, the proportions of pure-Ant and Ant-rich rosettes should be much lower in comparison with those of heteromeric rosettes consisting of Ant(4)-Nap(2) and Ant(3)-Nap(3). If so, then the following discussions could be misleading because the situations in the pure sample and the mixture (of Ant and Nap in different concentrations) are quite different:

(page 7, line 19) "The T_e of Ant at $c=1.0 \cdot 10^{-5}$ M(361K) is higher than that of Nap at $c=1.5 \cdot 10^{-5}$ M(353K). This result indicates that Ant nucleates first in a mixture at this ratio, which is consistent with the experimentally observed helical-linear-helical structure."

(page 11, line 20) "The low yield of the chimeric fibers and their predominant linear segments can be correlated with the smaller overlap of the temperature regime for polymerizing pure Ant and Nap: the molar fraction of aggregated Ant molecules already reaches 0.65 at the T_e (345 K) of Nap (Supplementary Fig. 6d)."

(page 12, line 4) "A larger overlap of the temperature regime of polymerization was achieved by increasing the concentration of Nap to e.g. $1.3 \cdot 10^{-5}$ M (1:1.3 mixture) and $1.5 \cdot 10^{-5}$ M (1:1.5 mixture), where the molar fraction of aggregated Ant molecules at the T_e (350 and 353 K) of Nap was reduced to 0.5 and 0.4, respectively (Supplementary Fig. 6d)."

Other comments:

(1) In Figure 3e (the case of narcissistic self-sorting), why do the green and black curves not overlap? It is also puzzling that the mixture (green curve) showed higher temperature for nucleation than pure Ant solution (blue curve).

(2) If Ant-rich rosettes first nucleate with a slower cooling rate (the upper scheme in Figure 3g), the initial spectral change at temperatures close to the nucleation temperature would be similar to that of pure Ant shown in Figure 2b. This would be more clearly shown in difference spectra because Nap and Ant have distinct isosbestic points in temperature-dependent absorption spectral changes (Figure 2g and 2h).

(3) As for the energy transfer experiments (page 9), excitation and absorption spectra of the chimeric structure and a mixture of preformed homopolymers should be compared.

(4) The authors determined photo-conversion yield in MCH based on the results obtained in CDCl₃ (Sup. Figure 19c,d). As the Ant have different absorption coefficient in monomeric and aggregated

form (Figure 2h), the result would not be reliable. Dilution of the MCH solution before/after the photo-dimerization with good solvents (such as CHCl₃) and measurement of absorption spectra at the disassembled states should provide better estimation of photo-conversion.

(5) The decrease in the Nap emission after the photodimerization was explained by the disconnection of the energy transfer pathway. But, it would be more likely due to the decrease in the concentration of the donor molecule (i.e., Ant).

Reviewer #2 (Remarks to the Author):

The authors describe the synthesis of the supramolecular block copolymers that bring about the topologically chimeric nanofibers with the linear and coiled nanostructures in a single chain. The barbiturate derivatives Nap and Ant were self-assembled in a nucleation-elongation regime. Below the transition temperatures, they formed the kinetically trapped unique nanostructures: Nap formed the helical coils published in ref. 49, and the linear fiber was from Ant. The narcissistic self-sorting behavior of a mixture of Nap and Ant was found in slow cooling. A fast cooling resulted in the co-assembly between Nap and Ant to form the gradient copolymers with the chimeric coiled and linear nanofibers. The co-assembled structure of Nap and Ant were studied using FRET technique. The excitation of Ant moiety in the co-assembled polymer resulted in the energy transfer from Ant to Nap. The Nap excimer emitted, while the assembly formed in a fast cooling was less emissive in this condition. Therefore, the co-assembled structure was built only in a fast cooling, which is consistent with the copolymer formation seen in AFM images. As shown in Figure 3, the kinetically controlled processes of the co-assembly of Nap and Ant were finely controlled. Overall pictures described in this manuscript seems to be convincing. This ms contains a portion of their previously published work in ref.49. I think of this ms might become publishable in Nature Communications after justifying the following points.

- The authors claim that the gradient copolymer drove the chimeric structures. But, the FRET experiments don't support the formation of the gradient polymer structure. How do the authors exclude a random copolymer structure in these experiments? The authors should provide further evidence to prove the formation of the gradient structure in the copolymer.
- The Ant emission didn't show any shift during the assembly. But I think of the electronic structure of the assemblies should be different from that of monomer, so the emission can be shifted upon the assembling process. The absorption changes of the Ant assembly should be provided. The authors should describe why the FL didn't shift at all. The author's comments for this are needed.
- The supramolecular polymerization is in a nucleation-elongation regime. I think of this should show a living nature. A degree of polymerization and PDI should be provided to support the living nature in polymerization.

Reviewer #3 (Remarks to the Author):

The manuscript "One-shot preparation of topologically chimeric nanofibers via a gradient supramolecular copolymerization" by Yagai and coworkers is a wonderful piece of work. Two almost similar supramolecular motifs, only differing in one benzene unit, self-sort when coassembled under thermodynamically controlled conditions, but coassemble in very interesting manner when the coassembly is done with temperature controlled cooling. In the latter case, the signatures of the individual components are found back in the coassembly, but the coassembly features both signatures blended together in a well structured manner. The paper is very well structured and

written and every question that comes up is clearly addressed and explained by the authors. The way the authors step by step unravel the details of the coassembly process is really nicely done and can serve as an inspiration to others. I find the work very interesting, especially the realisation of what kinetic control can bring in a supramolecular copolymerization. I have two really minor issues thae authors may want to address if they happen to have the experimental data available. These questions are born more out of curiosity, than that anything needs to be altered in the current version of the manuscript.

1. The authors say that the kinetically trapped coassemblies are very stable, even after keeping them for one week. However, what happens when keeping them even longer and at slightly elevated temperatures in solution. Does it then convert to the self-sorted system?
2. The authors have now applied MCH. In how far does the solvent structure affect the observed kinetic trapping of he co-assemblies? I can imagine that solvation also lays a role in the kinetics of the process. Did the authors ever tried another, maybe unbranched alkane??

Answer to the comments raised by reviewer 1 (Additions to the main text are highlighted yellow or blue)

Reviewer #1

Comments:

In this manuscript, the authors report formation of topologically unique 'chimeric' nanofiber. Such a complex structure has not been achieved before, and this beautiful demonstration will surely lead to design of nanostructures such as those of proteins as the authors aim at. The work deserves publication in Nature Communications. I hope that the authors address the following comments before publication.

Reply: Thank you very much for favorable comments. We are very happy to receive such a high praise. To improve the quality of our manuscript, we revised it according to the reviewer's comments as follows.

I agree that the hierarchical nature of supramolecular polymerization via the formation of hydrogen-bonded intermediate is crucial. I think the proposed mechanism is qualitatively correct, but as the authors admit by describing that "It should be noted that the respective segments are not purely composed of one monomer, but contain a minor amount of the co-monomer, as already demonstrated by the experiments for the largely biased mixtures (page 12)", the system is complex. The authors sought to quantitatively discuss the mechanism, but it has made discussion confusing (or misleading). I hope that the authors can estimate statistic proportions of each rosette (13 patterns) in the mixture before the nucleation. In the mixture, the proportions of pure-Ant and Ant-rich rosettes should be much lower in comparison with those of heteromeric rosettes consisting of Ant₍₄₎-Nap₍₂₎ and Ant₍₃₎-Nap₍₃₎. If so, then the following discussions could be misleading because the situations in the pure sample and the mixture (of Ant and Nap in different concentrations) are quite different:

(page 7, line 19) "The T_e of Ant at $c=1.0 \cdot 10^{-5}$ M(361K) is higher than that of Nap at $c=1.5 \cdot 10^{-5}$ M(353K). This result indicates that Ant nucleates first in a mixture at this ratio, which is consistent with the experimentally observed helical-linear-helical structure."

(page 11, line 20) "The low yield of the chimeric fibers and their predominant linear segments can be correlated with the smaller overlap of the temperature regime for polymerizing pure Ant and

Nap: the molar fraction of aggregated Ant molecules already reaches 0.65 at the Te (345 K) of Nap (Supplementary Fig. 6d)."

(page 12, line 4) "A larger overlap of the temperature regime of polymerization was achieved by increasing the concentration of Nap to e.g. 1.3×10^{-5} M (1:1.3 mixture) and 1.5×10^{-5} M (1:1.5 mixture), where the molar fraction of aggregated Ant molecules at the Te (350 and 353 K) of Nap was reduced to 0.5 and 0.4, respectively (Supplementary Fig. 6d)."

Reply: Thank you very much for the important comments. Importantly, before nucleation, **Ant** and **Nap** molecules form equilibrium mixture of diverse hydrogen-bonded species, not only rosettes. By forming rosettes, **Ant** and **Nap** can nucleate due to self-recognition process through π - π stacking. In other words, the formation of rosettes is driven not only by hydrogen-bonding but also π - π stacking interactions. Based on this mechanism, **Ant** and **Nap**-rich rosettes preferentially nucleate and elongate from the diverse hydrogen-bonded species even though their statistical concentrations are always very low. Accordingly, it is reasonable to discuss the temperature-dependent UV-vis data of the copolymerization processes on the basis of the corresponding homopolymerization processes at the same monomer concentrations. To avoid confusion and/or misleading of readers, we revised the related parts (yellow highlight in page 7).

Other comments:

(1) In Figure 3e (the case of narcissistic self-sorting), why do the green and black curves not overlap? It is also puzzling that the mixture (green curve) showed higher temperature for nucleation than pure Ant solution (blue curve).

Reply: Thank you very much for the valuable comments. The deviation of the experimental curve (green curve) and simulated curve (black curve) in a cooling rate of 0.1 K min^{-1} in the nucleation regime (Figure 3e) is most likely because a minor amount of **Nap** is taken in **Ant** rosettes to form **Ant**-rich rosettes (e.g., rosette composed of five **Ant** and one **Nap** molecules). If such **Ant**-rich rosettes are allowed to nucleate with **Ant** homomeric rosette, the nucleation should occur at a higher temperature in comparison with pure **Ant** because of apparent increase of the concentration of **Ant**, which is in line with the experimental data. Similarly, the larger ΔA of the experimental curve (green curve) than that of the simulated curve (black curve) in the elongation regime can be explained in line with the incorporation of a minor amount of **Ant** in **Nap** rosettes. The incorporated **Ant** molecules should also form slipped stacking as **Nap** molecules do, and may

contribute an increase of ΔA at 470 nm due to bathochromic shift. We therefore added this explanation in addition to revising the phrase “narcissistically self-sort” to “quasi-narcissistically self-sort into **Ant**-rich and **Nap**-rich fibers” (blue highlight in page 8-9).

(2) If **Ant**-rich rosettes first nucleate with a slower cooling rate (the upper scheme in Figure 3g), the initial spectral change at temperatures close to the nucleation temperature would be similar to that of pure **Ant** shown in Figure 2b. This would be more clearly shown in difference spectra because **Nap** and **Ant** have distinct isosbestic points in temperature-dependent absorption spectral changes (Figure 2g and 2h).

Reply: According to the reviewer’s suggestion, we have compared the initial absorption spectra in the “quasi-narcissistic” self-sorting system at 361 K (green curve) and the summation of the two homopolymers at 358 K (black curve) realized by 0.1-K min⁻¹ cooling. As shown by the spectra below, the two spectra are well overlapped, which indicates that **Ant**-rich rosettes first nucleate in the quasi-narcissistic self-sorting system. This clearly supports the absorbance change at 470 nm shown in Figure 3e. We revised the corresponding part in the manuscript (yellow highlight in page 9) and provided the data as Supplementary Figure 7.

New Supplementary Figure 7:

Supplementary Fig. 7. Comparison of the UV-vis absorption spectra of pure **Ant and a mixture with **Nap**.** UV-vis absorption spectrum for a 1:1.5 mixture of **Ant** and **Nap** at 361 K (green curve, [**Ant**] = 1.0×10^{-5} M; [**Nap**] = 1.5×10^{-5} M) in MCH obtained by cooling the solution from 373 to 361 K using a cooling rate of 0.1 K min⁻¹. The black curve corresponds to a reference curve obtained by a simple summation of the absorption spectra of the two

homoassemblies prepared by cooling each solution from 373 to 358 K using a cooling rate of 0.1 K min⁻¹.

(3) As for the energy transfer experiments (page 9), excitation and absorption spectra of the chimeric structure and a mixture of preformed homopolymers should be compared.

Reply: Thank you for your suggestion. Before the initial submission, we also discussed the use of excitation spectra to study FRET. However, as seen from the overlapped absorption spectra of **Nap** and **Ant** shown below (and also shown in Figure 4a), the absorption of **Nap** in the visible region entirely covers that of **Ant**. This indicates that we cannot discriminate whether the broad emission of **Nap** can be also obtained upon excitation of **Ant**. Therefore we would like to refrain from using excitation spectra to study FRET because complicated explanation might be necessary to argue for such a weak evidence.

(4) The authors determined photo-conversion yield in MCH based on the results obtained in CDCl₃ (Sup. Figure 19c,d). As the Ant have different absorption coefficient in monomeric and aggregated form (Figure 2h), the result would not be reliable. Dilution of the MCH solution before/after the photo-dimerization with good solvents (such as CHCl₃) and measurement of absorption spectra at the disassembled states should provide better estimation of photo-conversion.

Reply: Thank you very much for this valuable suggestion. According to the reviewer's suggestion, we have performed the following experiment. After UV-irradiation of the solution of **Ant** in MCH, the solvent was dried up by N₂ flow. The resulting film was dissolved in

chloroform and then the absorption spectrum was measured. The photo-conversion yield estimated from this experiment was 16%. Accordingly, we have revised the original value and the data in the revised Supplementary Figure 20c (blue highlight in page 13)

(5) The decrease in the Nap emission after the photodimerization was explained by the disconnection of the energy transfer pathway. But, it would be more likely due to the decrease in the concentration of the donor molecule (i.e., Ant).

Reply: Thank you for this important suggestion. In order to say that the decreased concentration of **Ant** was the cause of the decrease in the **Nap** emission after the photodimerization, we used the phrase “the disconnection of the energy transfer pathway”. This is therefore a mistake arising from our low expressiveness, and we revised the corresponding part (blue highlight in page 14).

Answer to the comments raised by reviewer 2 (Additions to the main text are highlighted green)

Reviewer #2

Comments:

The authors describe the synthesis of the supramolecular block copolymers that bring about the topologically chimeric nanofibers with the linear and coiled nanostructures in a single chain. The barbiturate derivatives Nap and Ant were self-assembled in a nucleation-elongation regime. Below the transition temperatures, they formed the kinetically trapped unique nanostructures: Nap formed the helical coils published in ref. 49, and the linear fiber was from Ant. The narcissistic self-sorting behavior of a mixture of Nap and Ant was found in slow cooling. A fast cooling resulted in the co-assembly between Nap and Ant to form the gradient copolymers with the chimeric coiled and linear nanofibers. The co-assembled structure of Nap and Ant were studied using FRET technique. The excitation of Ant moiety in the co-assembled polymer resulted in the energy transfer from Ant to Nap. The Nap excimer emitted, while the assembly formed in a fast cooling was less emissive in this condition. Therefore, the co-assembled structure was built only in a fast cooling, which is consistent with the copolymer formation seen in AFM images. As shown in Figure 3, the kinetically controlled processes of the co-assembly of Nap and Ant were finely controlled. Overall pictures described in this manuscript seems to be convincing. This ms contains a portion of their previously published work in ref.49. I think of this ms might become publishable in Nature Communications after justifying the following points.

Reply: We would like to thank the reviewer for favorable comments. To improve the quality of our manuscript, we have addressed the reviewer's comments as follows.

- The authors claim that the gradient copolymer drove the chimeric structures. But, the FRET experiments don't support the formation of the gradient polymer structure. How do the authors exclude a random copolymer structure in these experiments? The authors should provide further evidence to prove the formation of the gradient structure in the copolymer.

Reply: In the manuscript, we do not use the FRET phenomenon itself to support the formation of the gradient polymer structure although we believe that it is unequivocal evidence of 1D heterojunction between **Nap** and **Ant** domains. We believe that the temperature-dependent absorption change presented in Figure 3f could be an unequivocal evidence of the gradient fashion

of copolymerization, and strongly exclude the possibility of random copolymerization in addition to the “chimeric features” of our supramolecular copolymers shown in Figure 5. If the copolymerization undergoes in random fashion, the temperature-dependent changes in the absorption spectra do not show such a characteristic down–up trajectory upon cooling, and instead should show a simple down or up trajectory. The AFM images of chimeric fibers are proof that the present supramolecular copolymerization does not undergo in random fashion.

- The Ant emission didn't show any shift during the assembly. But I think of the electronic structure of the assemblies should be different from that of monomer, so the emission can be shifted upon the assembling process. The absorption changes of the Ant assembly should be provided. The authors should describe why the FL didn't shift at all. The author's comments for this are needed.

Reply: Thank you very much for suggesting this important point. Regarding the absorption spectra of **Ant**, we have already presented the data in Figure 2h, and discussed in Page 5 in the original manuscript. Unlike to **Nap** that shows a red-shifted absorption band compared to monomeric state (Figure 2g), **Ant** shows a hypochromic effect upon supramolecular polymerization. These contrasting absorption changes of **Ant** and **Nap** upon supramolecular polymerization indicate the different stacking mode of acene moieties. As shown by molecular modeling in Supplementary Figure 4, a largely overlapped stacking of the anthracene moieties was observed for **Ant**. This is in striking contrast to a largely slipped stacking of **Nap**. In line with these absorption and molecular modeling results, **Nap** molecule exhibits a new broad emission band upon supramolecular polymerization, which can be attributed to an excimer emission derived from largely slipped stacking of the naphthalene moieties (Supplementary Figures 4c and 8a). On the other hand, **Ant** molecule exhibits a simple increase of the locally excited (LE) state emission because it cannot form excimer (Supplementary Figures 4d and 8b). Probably the largely overlapped stacking of the anthracene moieties do not allow any excitonic interaction, and accordingly, enhancement of the LE emission occurred as a result of the suppression of bond twist in the excited state by aggregation. We therefore added this point in the revised manuscript (green highlight in page 9). We are grateful to the reviewer for suggesting this important point.

- The supramolecular polymerization is in a nucleation-elongation regime. I think of this should show a living nature. A degree of polymerization and PDI should be provided to support the living nature in polymerization.

Reply: Thank you very much for giving an interesting comment. As suggested by the reviewer, our system may have a living nature because **Nap** and **Ant** before nucleation randomly associate through hydrogen-bonding to form kinetic aggregates. We thus estimated a degree of polymerization (DP) and poly dispersity index (PDI) by analyzing the AFM images of the chimeric fibers. The DP and PDI values are summarized in the following table. The average DP of the linear segment (DP_{linear}) was estimated to be 5.5×10^2 , and the average DP of the coiled segments (DP_{coil}) was estimated to be 1.0×10^3 . The ratio of DP_{linear} and DP_{coil} is thus 1:1.8, which is comparable to the mixing ratio of **Ant** and **Nap** molecules (1:1.5). This result indicates that the major components of linear and coiled segments are **Ant** and **Nap**, respectively. On the other hand, the calculated PDI of chimeric fibers, linear and coiled segments are all in the range of 1.6–1.7, indicating polydisperse nature. In most cases of living supramolecular polymerization, for instance the pioneering work by Sugiyasu et al. (*Nature Chem.* **2014**, *6*, 188–195), the experiments have been performed under isothermal condition using metastable aggregates or dormant monomers that have definite lag time for spontaneous nucleation. On the other hand, we applied temperature-controlled supramolecular polymerization to kinetically achieve the gradient copolymerization. This typically makes lag time of metastable aggregates very short or negligible due to spontaneous polymerization upon continuous temperature descent. To avoid further complicated discussion, we refrain from discussing this topic. However, we would like to report DP values in the revised manuscript (green highlight in page 12, and Supplementary Table 1) because our present study is very “polymer-like”. Thank you very much again for giving us this fruitful advice.

	Chimeric fibers	Linear segments in chimeric fibers	Coiled segments in chimeric fibers
DP	1.6×10^3 ($l_{\text{av}} = 4.9 \times 10^2 \text{ nm}$) ^a	5.5×10^2 ($l_{\text{av}} = 2.0 \times 10^2 \text{ nm}$) ^a	1.0×10^2 ($l_{\text{av}} = 2.8 \times 10^2 \text{ nm}$) ^a
PDI	1.6	1.6	1.7

^a l_{av} : average length of chimeric fibers and each segment in the chimeric fibers.

Answer to the comments raised by reviewer 3

Reviewer #3

Comments:

The manuscript "One-shot preparation of topologically chimeric nanofibers via a gradient supramolecular copolymerization" by Yagai and coworkers is a wonderful piece of work. Two almost similar supramolecular motifs, only differing in one benzene unit, self-sort when coassembled under thermodynamically controlled conditions, but coassemble in very interesting manner when the coassembly is done with temperature controlled cooling. In the latter case, the signatures of the individual components are found back in the coassembly, but the coassembly features both signatures blended together in a well structured manner. The paper is very well structured and written and every question that comes up is clearly addressed and explained by the authors. The way the authors step by step unravel the details of the coassembly process is really nicely done and can serve as an inspiration to others. I find the work very interesting, especially the realisation of what kinetic control can bring in a supramolecular copolymerization. I have two really minor issues thae authors may want to address if they happen to have the experimental data available. These questions are born more out of curiosity, than that anything needs to be altered in the current version of the manuscript.

Reply: We would like to thank the reviewer for favorable comments. We are very happy to receive such a high praise.

1. The authors say that the kinetically trapped coassemblies are very stable, even after keeping them for one week. However, what happens when keeping them even longer and at slightly elevated temperatures in solution. Does it then convert to the selfsorted system?

Reply: When the solutions of the kinetically trapped coassemblies (i.e., chimeric fibers) were aged for one month at 20 °C, no significant topological change was observed for the chimeric structures. We have not examined aging of the chimeric fibers at slightly elevated temperature below the onset of dissociation (e.g., 30 °C). However, we believe that self-sorting of **Nap** and **Ant** does not occur since we have already confirmed that our supramolecular polymers are very static in terms of exchange dynamics by mixing experiments of helically coiled and randomly coiled fibers of **Nap** (*Sci. Adv.*, **2018**, 4, eaat8466).

2. The authors have now applied MCH. In how far does the solvent structure affect the observed kinetic trapping of the coassemblies? I can imagine that solvation also plays a role in the kinetics of the process. Did the authors ever try another, maybe unbranched alkane??

Reply: In our system, the selection of MCH with a high boiling point (101 °C) is essential for dissolution of **Ant** and **Nap** molecules to obtain monomeric states. Although some acyclic higher alkanes also have high boiling points, they are not good medium to realize monomeric states of **Ant** and **Nap** due to low solubility. We are now investigating the kinetic and thermodynamic aspects of supramolecular polymerization in cyclic and acyclic alkanes using some of our molecules having good solubility. For the use of aromatic co-solvents, such as toluene, we have not applied this system. However, it might be possible to control kinetics of supramolecular copolymerization. We will examine co-solvent system in near future. Thank you very much for pointing out an interesting point.

REVIEWERS' COMMENTS:

Reviewer #1 (Remarks to the Author):

The authors have carefully addressed the comments raised by the reviewers. The manuscript has been improved and become clearer. I think that in-depth FRET study is missing, but it is not a critical issue in this study. I recommend publication of this manuscript in Nature Communications.

Reviewer #2 (Remarks to the Author):

I am very much satisfied with the revision. Now the manuscript is acceptable as is.

Reviewer #3 (Remarks to the Author):

In the revised manuscript "One-shot preparation of topologically chimeric nanofibers via a gradient supramolecular copolymerization" by Yagai and coworkers, the questions raised by this and the other reviewers have been adequately addressed. The additions to the manuscript have made it even clearer and the data fully support the claims made, despite the fact that this is clearly a very complex system. I can recommend acceptance as is